# Aprotic Ionic Liquids: A Framework for Predicting Vaporization Thermodynamics

**DOI:** 10.3390/molecules27072321

**Published:** 2022-04-03

**Authors:** Sergey P. Verevkin, Dzmitry H. Zaitsau, Ralf Ludwig

**Affiliations:** 1Institut für Chemie, Abteilung für Physikalische Chemie, Universität Rostock, 18059 Rostock, Germany; dzmitry.zaitsau@uni-rostock.de (D.H.Z.); ralf.ludwig@uni-rostock.de (R.L.); 2Department Life, Light & Matter, University of Rostock, 18059 Rostock, Germany; 3Leibniz-Institut für Katalyse an der Universität Rostock e.V., 18059 Rostock, Germany

**Keywords:** ionic liquids, vapor pressure measurements, enthalpy of vaporization, structure–property relationships

## Abstract

Ionic liquids (ILs) are recognized as an environmentally friendly alternative to replacing volatile molecular solvents. Knowledge of vaporization thermodynamics is crucial for practical applications. The vaporization thermodynamics of five ionic liquids containing a pyridinium cation and the [NTf_2_] anion were studied using a quartz crystal microbalance. Vapor pressure-temperature dependences were used to derive the enthalpies of vaporization of these ionic liquids. Vaporization enthalpies of the pyridinium-based ionic liquids available in the literature were collected and uniformly adjusted to the reference temperature *T* = 298.15 K. The consistent sets of evaluated vaporization enthalpies were used to develop the “centerpiece”-based group-additivity method for predicting enthalpies of vaporization of ionic compounds. The general transferability of the contributions to the enthalpy of vaporization from the molecular liquids to the ionic liquids was established. A small, but not negligible correction term was supposed to reconcile the estimated results with the experiment. The corrected “centerpiece” approach was recommended to predict the vaporization enthalpies of ILs.

## 1. Introduction

Volatile *molecular* solvents are widely used in the chemical industry for extraction, recrystallization or as a reaction medium. The separation of solvents from chemicals is usually performed by evaporation or distillation [1,2,3,4,5,6,7]. It is inevitable that a significant amount of volatile solvents will be lost into the atmosphere and pollute nature. The extremely low-volatility *ionic* solvents, on the other hand, do not have this disadvantage and are considered to be a substitute for molecular solvents in many technical applications. Chemical processes are usually carried out at elevated temperatures where vapor pressures cannot be considered negligible. Therefore, a reliable knowledge of the vaporization thermodynamics is essential to avoid material losses or to reveal the limits of thermal stability [8,9,10].

Obtaining the proper vapor measurements of ionic liquids (ILs) are a challenging task because, at low temperatures, the vapor pressure is too low to be measured, but at high temperatures the decomposition processes can occur and falsify the result. For this reason, the experimental results for the vaporization thermodynamics of ionic liquids should be validated before they can be considered as reliable. [8,9,10] It is evident that a better understanding of transition enthalpies also helps to elucidate macroscopic fluid phenomena, and thus promote industrial applications.

Structure–property relationships are a very useful diagnostic tool to establish the internal consistency of available experimental data. They are suitable for predicting at least the level of the property being measured when the data is known for similarly shaped molecules [11].

Unfortunately, only very few vapor pressures and vaporization enthalpies are available for ionic liquids [12]. This work contributes with six new datasets on the vaporization thermodynamics of pyridinium-based *ionic* liquids with bis(trifluoromethylsulfonyl)imide anion (see Figure 1) and the investigation of the interplay of structure–property relationships in *ionic* liquids, and leads to the development of a new general approach to predict vaporization enthalpies of *ionic* liquids using the available knowledge on the vaporization thermodynamics of *molecular* liquids. In our previous studies, we found that ionic liquids containing the [NTf_2_] anion are best suited for vapor pressure measurements because of their remarkable thermal stability. The pyridinium-based ionic liquids are used as the first part to understand structure–property relationships in ionic liquids. The similar study of the imidazolium ILs is ongoing.

The main idea of this work is demonstrated in Figure 2. As a matter of fact, in our previous work, we showed that the general regularities revealed for *molecular* liquids can be transferred to *ionic* liquids [13].

Indeed, the vaporization enthalpies of, e.g., pyridine derivatives, can be reliably predicted with the help of a simple contribution, $\Delta_{l}^{g}H_{m}^{o}$**(H→****R_1_)**, representing the replacement of an H atom in pyridine with any substituents (see Figure 2a). It was found that, for pyridinium-based ionic liquids, the same numerical values for the contributions $\Delta_{l}^{g}H_{m}^{o}$**(H→****R_1_)** can be used to estimate their vaporization enthalpies $\Delta_{l}^{g}H_{m}^{o}$(298.15 K) as given in Figure 2b.

In this work, we ascertain and generalize this finding based on the available data on vaporization enthalpies of pyridinium ionic liquids with the help of complementary measurements of six new ionic liquids.

## 2. Materials and Methods

The samples of alkyl-pyridinium-based ionic liquids with the bis(trifluoromethylsulfonyl)imide anion of 99% purity were of commercial origin (see Appendix A) and were used as received. Before starting the experiment, however, a sample of an IL was placed in an open cavity of the thermostatted block and subjected to vacuum (10^−5^ Pa) for conditioning. The quartz crystal microbalance (QCM) [14] was used for vapor pressure measurements at different temperatures. A sample of an IL was placed in an open cavity (Langmuir evaporation) of the thermostatted block inside the set-up. The standard molar enthalpies of vaporization, $\Delta_{l}^{g}H_{m}^{o}$, were derived from the temperature dependences of vapor pressures. A concise description of the L-QCM (Langmuir quartz crystal microbalance) method and data treatment is presented in the Appendix A.

## 3. Results and Discussion

### 3.1. Experimental Vaporization Thermodynamics of Pyridinium Based ILs 

The original experimental vapor pressures of Ils at different temperatures are collected in Appendix A. They were used to derive the standard molar enthalpies of vaporization $\Delta_{l}^{g}H_{m}^{o}$(*T*_av_), which are referenced to the average temperatures *T*_av_. These results are shown in Table 1, column 5. For thermochemical calculations, the vaporization enthalpies are used to adjust to the reference temperature *T* = 298.15 K. The $\Delta_{l}^{g}H_{m}^{o}$(298.15 K) values are calculated according to the Kirchhoff’s equation:(1)$$\Delta_{l}^{g}H_{m}^{o}(298.15\;K)=\Delta_{l}^{g}H_{m}^{o}\left(T_{\mathrm{av}}\right)+\Delta C_{p,m}^{o}\times \;(T_{\mathrm{av}}-298.15\;K)$$

The value $\Delta_{l}^{g}C_{p,m}^{o}$ = $C_{p,m}^{o}$(g) − $C_{p,m}^{o}$(liq) is the difference between the molar heat capacities of the gaseous $C_{p,m}^{o}$(g) and the liquid phase $C_{p,m}^{o}$(liq), respectively. The required $\Delta_{l}^{g}C_{p,m}^{o}$ values are presented in Table 1, column 6.

The compilation of experimental thermodynamic data of pyridinium-based ILs measured using the L-QCM technique is presented in Table 1. The $\Delta_{l}^{g}H_{m}^{o}$(298.15 K)-values for methyl- and cyano-substituted pyridinium-based ILs were measured for the first time (except for [3-CN-1-C4-Py][NTf_2_] data reported in our previous work [13]). To investigate the structure–property relationships, we also compiled (see Table 2 and Table 3) the vaporization enthalpies of the pyridinium-based ILs connected to the [NTf_2_] anion available in the literature.

### 3.2. Comparison of the Vaporization Enthalpies Derived from the Theoretical and Empirical Methods

Taking into account the difficulties of experimental measurements of the extremely low vapor pressures, the vaporization enthalpies should be compared to results obtained from other methods (see Table 4 and Table 5).

#### 3.2.1. Molecular Dynamic (MD)

A number of different MD simulation methods were used to calculate the vaporization enthalpies of ionic liquids [20,21,22,23] with varying degrees of success (see Table 4). The General AMBER Force Field (GAFF) [20] failed to predict the enthalpy of the vaporization of [1-C_3_-Py][NTf_2_] properly. Additionally, the original CL&P FF method [21] heavily overestimates the vaporization enthalpy of [1-C_4_-Py][NTf_2_]. However, after the refinement of this method [21], an acceptable agreement with the experiment was achieved for [1-C_4_-Py][NTf_2_] (see Table 4). Borodin [22] used a version of the MD simulation package Lucretius for MD simulations, which includes many-body polarization simulations. His result for the vaporization enthalpy of [1-C_4_-Py][NTf_2_] is in excellent agreement with the experiment. The empirical force fields are usually parametrized with experimental thermodynamic and structural data. Hence, our new experimental results on vaporization enthalpies for pyridinium-based series can be used for the development, re-parametrization, and validation of modern MD methods [10,28,29].

#### 3.2.2. COSMO

The quantum-chemistry-based model, COSMO with modifications COSMO*therm* [23] and COSMO-RS [24], was used for the prediction of the thermochemical properties of ILs. However, as shown in Table 4, the $\Delta_{l}^{g}H_{m}^{o}$(298.15 K) values calculated by the original COSMO-RS [24] are systematically (of about 10 kJ·mol^−1^) higher, compared to the experimental result. In contrast, the result for [1-C_6_-Py][NTf_2_] predicted by the modified COSMO*therm* is in agreement with the experiment (see Table 4).

#### 3.2.3. CRDS Method

Gas-phase electronic absorption spectroscopy was successfully used for studies of 3-Me-1-ethylpyridinium and 1-butyl-3-methylimidazolium cations connected with the [NTf_2_] anion [27]. The vapor pressures at 400–430 K were derived from the measured absorbance. The vaporization enthalpy determined for [3-Me-C_2_-Py][NTf_2_] using the CRDS (cavity ring-down laser absorption spectroscopy) method is presented in Table 5 and is evidently too high.

#### 3.2.4. Gas Chromatographic Method (GC)

This method is based on the experimental infinite dilution activity coefficients $\gamma_{1}^{\infty }$ derived from the retention times of various solutes measured by gas chromatography using the IL as the solute [12]. The necessary details are presented in the Appendix A. Ban et al. [26] used this method and reported the vaporization enthalpies, $\Delta_{l}^{g}H_{m}^{o}$(298.15 K), of [1-C_n_-Py][NTf_2_] with alkyl chain *n* = 4,6 and 8), which are compiled in Table 4 and they appear to be reasonable, in comparison to other methods. We used the original data for $\gamma_{1}^{\infty }$ for [4-Me-1-C_4_-Py][NTf_2_], measured by Domanska and Marciniak [30], and derived, $\Delta_{l}^{g}H_{m}^{o}$(298.15 K) = 135.7 ± 3.0 kJ⋅mol^−1^ (see Table 5), which meets the level expectation.

#### 3.2.5. Empirical Model

In an empirical approach developed by Licence and Jones [25], the $\Delta_{l}^{g}H_{m}^{o}$(298.15 K)-value is decomposed into the Coulombic and van der Waals contributions from the cation and anion. Unfortunately, a very limited experimental data set on vaporization enthalpies was used for the parameterization of this approach. As can be seen from Table 4, the vaporization enthalpy $\Delta_{l}^{g}H_{m}^{o}$(298.15 K) = 154 kJ·mol^−1^ estimated by this model for [1-C_4_-Py][NTf_2_] is practically equal to $\Delta_{l}^{g}H_{m}^{o}$(298.15 K) = 153 kJ·mol^−1^ estimated for [1-C_6_-Py][NTf_2_], in contrast to the established growth trend of the vaporization enthalpy, with increasing chain-length dependence within the homologous series. Apparently, this method needs further development, but unfortunately no update since 2014 has been found in the literature. 

### 3.3. Validation of the Vaporization Enthalpies

The comparison of the experimental enthalpies of vaporization with those derived from theoretical and empirical methods of the previous section was not sufficient to validate the available enthalpies of vaporization of the pyridinium-based ILs. The consistency of the complete data set could be checked using structure–property correlations, e.g., chain-length dependence, or using the correlation between the vaporization enthalpy and surface tension of the ILs.

#### 3.3.1. Structure–Property Correlations: Chain-Length Dependence

The linear correlation of $\Delta_{l}^{g}H_{m}^{o}$(298.15 K) values with the number of carbon atoms in the alkyl chain within the homologue series of ionic liquids is a well-established phenomenon, e.g., for the series [C_n_mim][NTf_2_] [31] or [N(R)_4_][NTf_2_] [32]. We also correlated the $\Delta_{l}^{g}H_{m}^{o}$(298.15 K) values for the [1-C_n_-Py][NTf_2_] series (evaluated in Table 2) with the number of carbon atoms, *n*, in the alkyl chain attached to the cation nitrogen atom. The following correlation was obtained (see Appendix A):(2)$$\Delta_{l}^{g}H_{m}^{o}(298.15\;K)/\;\mathrm{kJ}\cdot {\mathrm{mol}}^{-1}=124.2+3.60\;\times \;n(\mathrm{with}\;R^{2}=0.9958)$$

The relatively high correlation coefficient *R*^2^ is evidence of a good consistency of experimental data approximated by Equation (2). 

The surface tension *σ*_298_ (surface tension at the reference temperature *T* = 298.15 K) as a thermophysical property is directly related to the vaporization enthalpy $\Delta_{l}^{g}H_{m}^{o}$(298.15 K). Is the chain-length dependence of *σ*_298_ linear for pyridinium ionic liquids [1-C_n_-Py][NTf_2_]? The compilation of the experimental *σ*_298_ values available in the literature is presented in Table 6. 

Using this data, a robust linear correlation with the alkyl chain length attached to the N atom of the cation was established according to Equation (3):

*σ*_298_(est) = −1.37 × *N*_C_ +40.0 (with *R*^2^ = 0.980)(3)

This correlation we used to estimate the *σ*_298_(est) values for [1-C_5_-Py][NTf_2_] and [1-C_8_-Py][NTf_2_] is required for the correlation with the vaporization enthalpies in Section 3.3.2.

#### 3.3.2. Correlation of the Vaporization Enthalpies with the Surface Tension

In this work, we correlated $\Delta_{l}^{g}H_{m}^{o}$(298.15 K) for the [1-C_n_-Py][NTf_2_] series with the surface tensions *σ*_298_ from Table 7, column 2. The results are presented in Table 7, column 4.

A good linear correlation has been found to be:(4)$$\Delta_{l}^{g}H_{m}^{o}(298.15\;K)/\mathrm{kJ}\cdot {\mathrm{mol}}^{-1}=228.9-2.62\;\times \;\sigma_{298}\;(\mathrm{with}\;R^{2}=0.995)$$
for the [1-C_n_-Py][NTf_2_] series. As shown in Table 7, the differences between the experimental and estimated values do not exceed 1 kJ·mol^−1^, demonstrating the consistency of the data set of the unsubstituted pyridinium-based ILs.

Can we also use *σ*_298_ values to prove the consistency of the evaporation data for Me- and CN-substituted pyridinium ILs? To test this, we carefully collected the surface tension data of methyl-substituted (see Table 8) and cyano-substituted (see Table 9) pyridinium-based ILs and correlated these values with the vaporization enthalpies evaluated in Table 1 and Table 2.

The following linear correlations were established:(5)$$\mathrm{for}\;[\mathrm{Me}\text{-}1\text{-}C_{n}\text{-}\mathrm{Py}][{\mathrm{NTf}}_{2}]:\;\Delta_{l}^{g}H_{m}^{o}(298.15\;K)=-2.72\;\times \;\sigma_{298}(\exp{)}^{\;}+232.1\;(\mathrm{with}\;R^{2}=0.980)$$
(6)$$\mathrm{for}\;[\mathrm{CN}\text{-}1\text{-}C_{n}\text{-}\mathrm{Py}][{\mathrm{NTf}}_{2}]:\;\Delta_{l}^{g}H_{m}^{o}(298.15\;K)=-4.16\;\times \;\sigma_{298}(\exp{)}^{\;}+284.5\;(\mathrm{with}\;R^{2}=0.986)$$

As shown in Table 7 and Table 8, the differences between the experimental and estimated values are mostly below 1 kJ·mol^−1^, which also demonstrates the sufficient consistency of the data set of the substituted pyridinium-based ILs evaluated in this work. This dataset can now be used to develop a methodology to predict the enthalpies of ILs, which are difficult to access experimentally.

### 3.4. Group Additivity to Predict the Vaporization Enthalpies of Ionic Liquids Using Contributions from Molecular Liquids

The enthalpy of vaporization is generally a measure of the intensity of intermolecular interactions that hold molecules together in the liquid state. When Van der Waals forces and hydrogen bonding prevail over molecular liquids, the additional strong Coulomb interactions are specific to ionic liquids. This makes the ionic liquids (or molten salts) extremely low volatile, with vaporization enthalpies between 120 and 180 kJ⋅mol^−1^ [31]. Group additivity (GA) methods are successfully used to predict vaporization enthalpies of molecular liquids. In conventional GA methods, the vaporization enthalpies of molecules are split up into smallest possible groups in order to obtain well-defined contributions. The prediction is then based on the idea of “LEGO^®^ bricks”, where the energetics of the molecule of interest are collected from the appropriate type and number of bricks. A comprehensive system of group contributions (or increments) covers the major classes of organic compounds [39]. Using the same method for ionic liquids composed of large organic cations and large organic/inorganic anions is impractical, due to too many “bricks” and a very limited amount of available experimental enthalpies of vaporization. To overcome these limitations, we developed a general approach to estimate the vaporization enthalpies based on a so-called “centerpiece” molecule [40,41]. This approach is closely related to the broadly used group additivity (GA) methods [39,42]. The idea of the “centerpiece” approach is to start the prediction with a potentially large “core” molecule that can generally mimic the structure of the molecule of interest, but, at the same time, must has a reliable enthalpy of vaporization. The ionic liquids are predestined for such an approach. The visualization of the “centerpiece” approach for R-substituted [C_2_-Py][NTf_2_] ionic liquid is presented in Figure 3 as an example.

Indeed, [1-C_2_-Py][NTf_2_] as the “centerpiece” model already bears the main energetic contributions to the vaporization enthalpy, due to the sum of the Coulomb and Van der Waals interactions present in the liquid phase. Such a bulk contribution is unique to [1-C_2_-Py][NTf_2_] as the “centerpiece” molecule and can hardly be captured by any other method. This special feature of the “centerpiece” approach significantly increases the reliability of the property prediction for similarly shaped molecules, e.g., [R-1-C_2_-Py][NTf_2_] (see Figure 3, right), where substituent (with contribution $\Delta_{l}^{g}H_{m}^{o}$**(H→****R)** to the vaporization enthalpy) is simply attached to the aromatic ring of the cation (see Figure 3, left),. For this reason, the $\Delta_{l}^{g}H_{m}^{o}$(298.15 K) estimated using the “centerpiece” approach for the [R-1-C_2_-Py][NTf_2_] ionic liquids can be ad hoc regarded as reliable, since the “main” contribution to the vaporization energetics from the [1-C_2_-Py][NTf_2_] is already well established and consistent with the results available for the [1-C_n_-Py][NTf_2_] series (R = alkyl with n= 1-6) [15]. It is quite evident that the contributions, $\Delta_{l}^{g}H_{m}^{o}$**(H→****R)**, for the “appending” to the [1-C_2_-Py][NTf_2_] “centerpiece” are comparatively small and affordable for a reliable assessment. As a matter of fact, in our recent work [13], we have shown that the $\Delta_{l}^{g}H_{m}^{o}$**(H→****Me)** and $\Delta_{l}^{g}H_{m}^{o}$**(H→****CN)** contributions derived from the *molecular liquids* (substituted pyridines and quinolines) are generally transferrable to *ionic liquids*. The numerical values for these contributions are presented in Table 10.

Let us consider the prediction of the vaporization enthalpy for [3-Me-1-C_3_-Py][NTf_2_] as an example (see Table 11). The [1-C_3_-Py][NTf_2_] with $\Delta_{l}^{g}H_{m}^{o}$(298.15 K) = 135.4 ± 1.5 kJ·mol^−1^ was used as the “centerpiece”. Contribution $\Delta_{l}^{g}H_{m}^{o}$**(H→****Me)** = 4.4 ± 0.3 kJ·mol^−1^ from Table 9 was appended to the aromatic ring in the three position to construct the desired IL. The resulting value $\Delta_{l}^{g}H_{m}^{o}$_(CP)_ = 138.9 ± 1.6 kJ·mol^−1^ is in fair agreement with the experimental result from the QCM study $\Delta_{l}^{g}H_{m}^{o}$_(exp)_ = 132.4 ± 1.6 kJ·mol^−1^. Similarly, we used the “centerpiece” approach to estimate the $\Delta_{l}^{g}H_{m}^{o}$_(CP)_ values for the collection of the ionic liquids (see Table 11), where reliable experimental vaporization enthalpies were available.

Even a quick look at the results presented in Table 11 can reveal that the “centerpiece” approach of about 5 kJ·mol^−1^ systematically overestimates the vaporization enthalpies, if we directly take the $\Delta_{l}^{g}H_{m}^{o}$**(H→****R)** contributions from molecular liquids to the ionic liquids. It is thus evident that the overestimation is quite independent of the type and position of the substituent on the aromatic ring. Moreover, two ionic liquids, [3-Me-1-C_4_-Py][BF_4_] and [3-Me-1-C_4_-Py][BF_4_], with the [BF_4_] anion of a totally different nature, also show the same trend as the [NTf_2_] ionic liquids (see the final two lines in Table 11). One of the plausible explanations for this observation is that the $\Delta_{l}^{g}H_{m}^{o}$**(H→****R)** contributions are derived from molecular liquids. For vaporizing molecular liquids, all types of interaction need to be overcome for bringing the monomer molecules into the gas phase. This situation is different for ionic liquids, which evaporate as ion pairs. Thus, an attractive cation–anion Coulomb interaction, hydrogen bonding between both ions and, in particular, a dispersion interaction within the ion pair are taken into the gas phase. The overestimation of the vaporization enthalpies in the order of 5 kJ·mol^−1^ by applying the “centerpiece” is in the order of the derived dispersion energies between the cations and anions in an ion pair [44,45,46,47,48]. Indeed, such a contribution should be more or less constant for all types of ionic liquids, and only marginally dependent on the nature of the cation and anion, as demonstrated in the present paper.

Admittedly, these forces play only a subordinate role in ionic liquids. Thus, a direct transfer of the $\Delta_{l}^{g}H_{m}^{o}$**(H→****R)** contributions from the molecular to the ionic liquids evidently requires a systematic correction. Considering the common nature of the systematics observed for the ILs in Table 11, we propose the application of the “centerpiece” approach to ILs, along with the correction term Δ = (−4.9 ± 0.8) kJ·mol^−1^ (see Table 11), to predict the $\Delta_{l}^{g}H_{m}^{o}$(298.15 K) values of the broad range of ionic liquids using a comprehensive collection of $\Delta_{l}^{g}H_{m}^{o}$**(H→****R)** contributions available from molecular liquids.

Such a straightforward procedure facilitates a rapid diagnostic of the experimental or theoretical vaporization enthalpies already available in the literature. For example, in Table 5, we collect the results for five ionic liquids of the [Alkyl-1-C_3_-Py][NTf_2_] series. The $\Delta_{l}^{g}H_{m}^{o}$(298.15 K) values for this series were of a different quality. Nonetheless, our calculations using the “corrected-centerpiece” approach allows for a reliable estimate of the “expected” level of vaporization enthalpy for each species in Table 5, to detect the “sick” data and help to improve the experimental and theoretical methods. A prime example to support this idea is the calculation for [3-Me-1-C_2_-Py][NTf_2_] (see Table 5, line 3 and Appendix A). The “corrected-centerpiece” result $\Delta_{l}^{g}H_{m}^{o}$(298.15 K) = 131.2 ± 1.6 kJ·mol^−1^ makes it clear that the $\Delta_{l}^{g}H_{m}^{o}$(298.15 K) = 172 ± 35 kJ·mol^−1^ measured by the CRDS method is definitely an error, even taking into account the extremely significant uncertainties. However, by having such a convenient tool as the “corrected-centerpiece” approach, it might be possible to improve the measuring technique, provided that the level of the “expected” vaporization enthalpy is preliminarily assessed.

## 4. Conclusions

The structure–property correlations have proven to be a useful diagnostic tool for predicting the vaporization enthalpies for ILs. The general transferability of the group contributions derived from *molecular* liquids to estimate the vaporization enthalpies of *ionic* liquids was demonstrated. It was shown that, with the “centerpiece” approach, it is possible to estimate the appropriate level of vaporization enthalpy. The further refinement of this approach with the small, but not negligible, correction term has helped brought the estimated results into agreement with the experiment. The corrected “centerpiece” approach was recommended to predict the vaporization enthalpies of ILs. The application of this approach to imidazolium-based ILs will be explored in the upcoming studies.

## Figures and Tables

**Figure 1 molecules-27-02321-f001:**
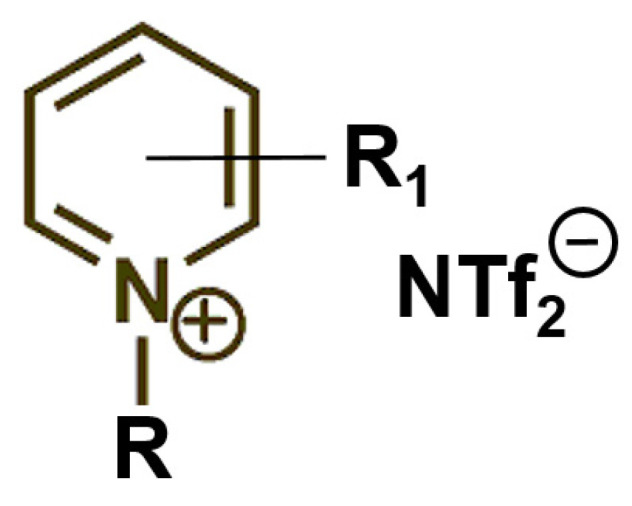
Structures of pyridinium-based ionic liquids studied in this work using a quartz crystal microbalance (QCM) with R = alkyl chain C_3_, C_4_, C_6_, and C_8_ and R_1_ = Me and CN. For brevity, the cations of ILs measured in this work are named as follows: [1-C_8_-Py] for of 1-octyl-pyridinium; [3-Me-1-C_3_-Py] for of 3-methyl-1-propyl-pyridinium; [3-CN-1-C_6_-Py] for 3-cyano-1-hexyl-pyridinium; [4-CN-1-C_6_-Py] for 4-cyano-1-hexyl-pyridinium; and [3-CN-1-C_8_-Py] for the 3-cyano-1-octyl-pyridinium cation connected with the bis(trifluoromethylsulfonyl)imide anion (abbreviation: [NTf_2_]).

**Figure 2 molecules-27-02321-f002:**
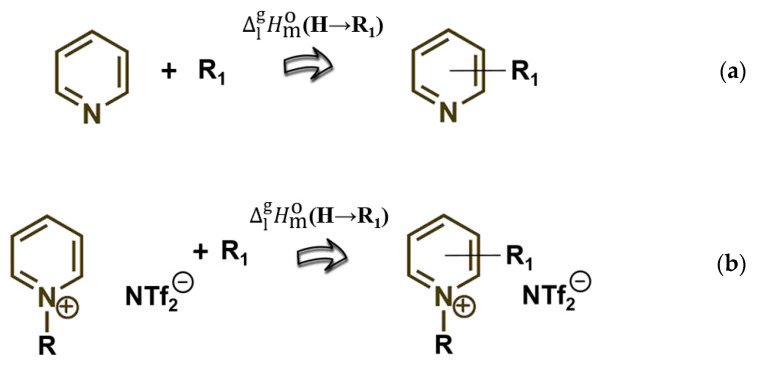
Assessment of the enthalpy of vaporization $\Delta_{l}^{g}H_{m}^{o}$(298.15 K) in *molecula*r (**a**) and in *ionic* liquids (**b**).

**Figure 3 molecules-27-02321-f003:**
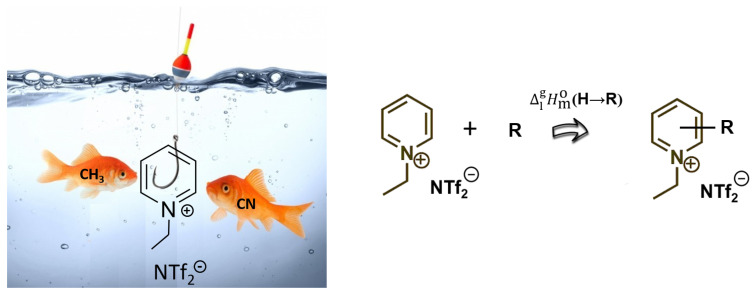
The visualization of the “centerpiece” approach for the [1-C_2_-Py][NTf_2_] substituted with methyl or cyano substituent (**left**). Estimation of $\Delta_{l}^{g}H_{m}^{o}$(298.15 K) values for [R-1-C_2_-Py][NTf_2_] (**right**).

**Table 1 molecules-27-02321-t001:** Results of L-QCM studies of pyridinium-based ionic liquids (in kJ·mol^−1^) ^a^.

Ionic Liquids	*T* _range_	*T* _av_	$\Delta_{l}^{g}G_{m}^{o}\left(T_{\mathrm{av}}\right)$	$\Delta_{l}^{g}H_{m}^{o}\left(T_{\mathrm{av}}\right){\;}^{b}$	$\Delta C_{p,m}^{o}{\;}^{c}$	$\Delta_{l}^{g}H_{m}^{o}\left(298.15\;K\right){\;}^{d}$
	K	K	kJ·mol^−1^	kJ·mol^−1^	J·mol^−1^·K^−1^	kJ·mol^−1^
[1-C_8_-Py][NTf_2_]	383.2–430.4	406.2	76.5 ± 1.5	142.4 ± 1.0	−100	153.2 ± 2.4
[3-Me-1-C_3_-Py][NTf_2_]	357.9–407.5	385.2	71.7 ± 1.5	126.3 ± 1.0	−70	132.4 ± 1.6
[3-CN-1-C_4_-Py][NTf_2_] ^e^	400.3–448.1	424.8	78.5 ± 1.5	141.8 ± 1.0	−75	151.2 ± 2.1
[3-CN-1-C_6_-Py][NTf_2_]	402.9–450.6	426.2	79.6 ± 1.5	151.3 ± 1.0	−84	162.1 ± 2.4
[4-CN-1-C_6_-Py][NTf_2_]	405.4–448.1	427.5	78.3 ± 1.5	147.9 ± 1.0	−81	158.4 ± 2.3
[3-CN-1-C_8_-Py][NTf_2_]	407.9–455.7	434.8	79.7 ± 1.5	152.9 ± 1.0	−91	165.3 ± 2.7

^a^ Uncertainties of vaporization enthalpy ($\Delta_{l}^{g}H_{m}^{o}$) and Gibbs free energy of vaporization ($\Delta_{l}^{g}G_{m}^{o}$) are the expanded uncertainties (0.95 level of confidence, k = 2). ^b^ Vaporization enthalpy measured in the specified temperature range (see Appendix A) and referenced to the average temperature $T_{\mathrm{av}}$. ^c^ The heat capacity differences were derived from an empirical equation: $\Delta C_{p,m}^{o}$ = −0.126 × $C_{p,m}^{o}\left(\mathrm{liq}\right)$
_exp_ − 1.5 (with *R*^2^ = 0.987). The heat capacity values $C_{p,m}^{o}\left(\mathrm{liq}\right)$
_exp_ are compiled in Appendix A. ^d^ Vaporization enthalpies $\Delta_{l}^{g}H_{m}^{o}\left(T_{\mathrm{av}}\right)$ were treated using Equation (1), with the help of heat capacity differences from column 5 to evaluate the enthalpy of vaporization at 298.15 K. The final uncertainties of vaporization enthalpy are expanded, taking into account the uncertainty of the heat capacity difference $\Delta_{l}^{g}C_{p,m}^{o}$ assigned to be of ± 20 J·K^−1^·mol^−1^. ^e^ From Ref. [13].

**Table 2 molecules-27-02321-t002:** Compilation of the enthalpies of vaporization $\Delta_{l}^{g}H_{m}^{o}$ for [1-C_n_-Py][NTf_2_] available in the literature.

IL	M ^a^	*T* _av_	$\Delta_{l}^{g}H_{m}^{o}\left(T_{\mathrm{av}}\right){\;}^{b}$	$\Delta C_{p,m}^{o}{\;}^{c}$	$\Delta_{l}^{g}H_{m}^{o}\left(298.15\;K\right){\;}^{d}$	Ref.
		*K*	kJ·mol^−1^	J·mol^−1^·K^−1^	kJ·mol^−1^	
[1-C_2_-Py][NTf_2_]	L-QCM	400.6	125.3 ± 1.0	−61	131.5 ± 1.6	[15]
	K-QCM	498.6	120.1 ± 1.2		132.4 ± 2.8	[16]
					**131.7 ± 1.4** ^e^	average
[1-C_3_-Py][NTf_2_]	L-QCM	398.2	128.0 ± 1.0	−66	134.6 ± 1.7	[15]
	K-QCM	504.5	124.1 ± 1.0		137.7 ± 2.9	[16]
					**135.4 ± 1.5** ^e^	average
[1-C_4_-Py][NTf_2_]	L-QCM	399.5	131.1 ± 1.0	−70	138.2 ± 1.7	[15]
		553.0	119.8 ± 2.2		137.6 ± 4.2	[17]
	K-QCM	506.8	121.9 ± 1.7		136.5 ± 3.4	[16]
					**137.8 ± 1.4** ^e^	average
[1-C_5_-Py][NTf_2_]	L-QCM	400.6	134.2 ± 1.0	−73	141.7 ± 1.8	[15]
[1-C_6_-Py][NTf_2_]	L-QCM	405.7	137.3 ± 1.0	−77	145.6 ± 1.9	[15]
	TPD	440.0	138.6 ± 4.0	−77	149.5 ± 3.0	[18]
					**146.1 ± 1.8** ^e^	average
[1-C_8_-Py][NTf_2_]	L-QCM	406.2	142.4 ± 1.0	−100	153.2 ± 2.4	Table 1

^a^ Method: K-QCM—Knudsen effusion cell combined with a quartz crystal microbalance; L-QCM—Langmuir evaporation from the open surface combined with the quartz crystal microbalance; TPD—temperature-programed desorption line-of-sight mass spectrometry at a ultra-high vacuum. ^b^ Vaporization enthalpies measured in the specified temperature range and referenced to the average temperature $T_{\mathrm{av}}$. ^c^ The heat capacity differences were derived in our previous work [15] from the experimental volumetric properties. ^d^ Vaporization enthalpies $\Delta_{l}^{g}H_{m}^{o}\left(T_{\mathrm{av}}\right)$ were treated using Equation (1), with help of the heat capacity differences from column 5 to evaluate the enthalpies of vaporization at 298.15 K. The final uncertainties of vaporization enthalpies are expanded, taking into account the uncertainty of the heat capacity difference $\Delta_{l}^{g}C_{p,m}^{o}$ assigned to be of ± 20 J·K^−1^·mol^−1^. ^e^ Weighted mean value. Values in bold are recommended for further thermochemical calculations. Uncertainty of the vaporization enthalpy *U*($\Delta_{l}^{g}H_{m}^{o}$) is the expanded uncertainty (0.95 level of confidence, k = 2).

**Table 3 molecules-27-02321-t003:** Compilation of the enthalpies of vaporization $\Delta_{l}^{g}H_{m}^{o}$ for [2-Et-C_n_-Py][NTf_2_] evaluated in this work from the data available in the literature [19].

Ionic Liquid	*T* _av_ ^a^	$\Delta_{l}^{g}H_{m}^{o}\left(T_{\mathrm{av}}\right){\;}^{b}$	$\Delta C_{p,m}^{o}{\;}^{c}$	$\Delta_{l}^{g}H_{m}^{o}\left(298.15\;K\right){\;}^{d}$
	K	kJ·mol^−1^	J·mol^−1^·K^−1^	kJ·mol^−1^
[2-Et-1-C_2_-Py][NTf_2_]	508.0	124.6 ± 1.4	−73	139.9 ± 3.4
[2-Et-1-C_3_-Py][NTf_2_]	510.4	121.0 ± 0.8	−76	137.2 ± 3.3
[2-Et-1-C_4_-Py][NTf_2_]	503.0	122.3 ± 0.6	−80	138.7 ± 3.3
[2-Et-1-C_5_-Py][NTf_2_]	510.5	127.3 ± 1.4	−84	145.1 ± 3.8
[2-Et-1-C_6_-Py][NTf_2_]	505.5	128.3 ± 0.6	−88	146.5 ± 3.7
[2-Et-1-C_7_-Py][NTf_2_]	508.2	131.4 ± 2.8	−92	150.7 ± 4.8
[2-Et-1-C_8_-Py][NTf_2_]	505.5	138.5 ± 1.7	−96	158.4 ± 4.3
[2-Et-1-C_9_-Py][NTf_2_]	522.8	139.7 ± 1.3	−100	162.1 ± 4.7
[2-Et-1-C_10_-Py][NTf_2_]	520.4	144.5 ± 1.6	−104	167.5 ± 4.9

^a^ Average temperature of the K-QCM experiments. ^b^ Vaporization enthalpies measured [19] in the specified temperature range and referenced to the average temperature $T_{\mathrm{av}}$. ^c^ The heat capacity differences were derived from an empirical equation: $\Delta C_{p,m}^{o}$ = −0.126 × $C_{p,m}^{o}\left(\mathrm{liq}\right)$
_exp_ − 1.5 (with *R*^2^ = 0.987). The heat capacity values $C_{p,m}^{o}\left(\mathrm{liq}\right)$
_exp_ are compiled in Appendix A. ^d^ Vaporization enthalpies $\Delta_{l}^{g}H_{m}^{o}\left(T_{\mathrm{av}}\right)$ were treated using Equation (1), with the help of the heat capacity differences from column 5 to evaluate the enthalpies of vaporization at 298.15 K. The final uncertainties of the vaporization enthalpy are expanded, taking into account the uncertainty of the heat capacity difference $\Delta_{l}^{g}C_{p,m}^{o}$ assigned to be of ± 20 J·K^−1^·mol^−1^.

**Table 4 molecules-27-02321-t004:** Comparison of the experimental and theoretical vaporization enthalpies $\Delta_{l}^{g}H_{m}^{o}$(298.15 K) of [1-C_n_Py][NTf_2_] (in kJ⋅mol^−1^).

Method	[1-C_2_Py]	[1-C_3_Py]	[1-C_4_Py]	[1-C_6_Py]	[1-C_8_Py]
GAFF [20]	-	125.0	-	-	-
CL&P FF original [21]	-	-	167.0	-	-
CL&P FF refined [21]	-	-	141.0	-	-
MD [22]	-	-	137.3	-	-
COSMO-therm [23]	-	-	-	142.0	-
COSMO-RS [24]	143.9 ± 10	143.1 ± 10	145.8 ± 10	-	-
Empiric [25]	-	-	154.0	153.0	-
$\gamma_{1}^{\infty }$ based (see text) [26]	-	-	139.0 ± 4.2	147.5 ± 4.4	153.6 ± 4.6
Experiment ^a^	**131.7 ± 1.4**	**135.4 ± 1.9**	**137.8 ± 1.4**	**146.1 ± 1.8**	**153.2 ± 2.4**

^a^ Experimental data on $\Delta_{l}^{g}H_{m}^{o}$(298.15 K, [1-C_n_Py][NTf_2_]) were obtained from Table 2. The extended uncertainty with *k* = 2 and confidence level 0.95 is presented.

**Table 5 molecules-27-02321-t005:** Comparison of the experimental, empirical, and theoretical vaporization enthalpies $\Delta_{l}^{g}H_{m}^{o}$(298.15 K) of [Alkyl-1-C_n_-Py][NTf_2_] (in kJ⋅mol^−1^).

Method	Method	$\Delta_{l}^{g}H_{m}^{o}(T,K)$	Ref.
[3-Me-1-C_2_-Py][NTf_2_]	CRDS ^a^	172 ± 35	[27]
	additivity	131.2 ± 1.6	Appendix A
[2-Et-1-C_2_-Py][NTf_2_]	COSMO-RS	143.2 ± 10	[24]
	K-QCM	139.9 ± 3.4	Table 3
	additivity	132.5 ± 1.6	Appendix A
[3-Me-1-C_3_-Py][NTf_2_]	COSMO-RS	138.6 ± 10	[24]
	additivity	134.9 ± 1.7	Appendix A
	L-QCM	**132.4 ± 1.6**	Table 1
[4-Me-1-C_3_-Py][NTf_2_]	COSMO-RS	143.4 ± 10	[24]
	additivity	135.2 ± 1.7	Appendix A
[4-Me-1-C_4_-Py][NTf_2_]	$\gamma_{1}^{\infty }$ based	135.7 ± 3.0	Appendix A
	additivity	137.6 ± 1.6	Appendix A

^a^ Measured by CRDS (cavity ring-down laser absorption spectroscopy). The experimental value $\Delta_{l}^{g}H_{m}^{o}$ (419 K) = 162 ± 35 kJ⋅mol^−1^ [27] was adjusted to the reference temperature *T* = 298.15 K, with the help of $\Delta C_{p,m}^{o}$ = −85 J·mol^−1^·K^−1^, derived as shown in Appendix A.

**Table 6 molecules-27-02321-t006:** Surface tension, *σ*_298_(exp), and chain-length dependence for the [1-C_n_-Py][NTf_2_] series (in mN⋅m^−1^).

Ionic Liquid	*N_C_*	*σ*_298_(exp)	*σ*_298_(est) ^a^	Δ ^b^
[1-C_2_-Py][NTf_2_]	2	37.4 [33]	37.2	0.2
[1-C_3_-Py][NTf_2_]	3	35.4 [34]	35.9	−0.5
[1-C_4_-Py][NTf_2_]	4	34.8 [35]	34.5	0.3
[1-C_5_-Py][NTf_2_]	5	-	33.1	-
[1-C_6_-Py][NTf_2_]	6	31.7 [36]	31.7	0.0
[1-C_8_-Py][NTf_2_]	8	-	29.0	-

^a^ Estimated from the chain-length dependence according to Equation (3). ^b^ The difference between columns 3 and 4.

**Table 7 molecules-27-02321-t007:** Surface tension *σ*_298_ for [1-C_n_-Py][NTf_2_] series available in the literature and the correlation of the vaporization enthalpies $\Delta_{l}^{g}H_{m}^{o}$(298.15 K) with the surface tension.

Ionic Liquid	*σ* _298_	$\Delta_{l}^{g}H_{m}^{o}(298.15\;K{)}_{\exp}{\;}^{a}$	$\Delta_{l}^{g}H_{m}^{o}(298.15\;K{)}_{\mathrm{calc}}{\;}^{b}$	Δ ^c^
	mN⋅m^−1^	kJ⋅mol^−1^	kJ⋅mol^−1^	
[1-C_2_-Py][NTf_2_]	37.4 [33]	131.7 ± 1.4	131.0	0.7
[1-C_3_-Py][NTf_2_]	35.4 [34]	135.4 ± 1.5	136.2	−0.8
[1-C_4_-Py][NTf_2_]	34.8 [35]	137.8 ± 1.4	137.8	0.0
[1-C_5_-Py][NTf_2_]	*33.1* ^d^	141.7 ± 1.8	142.2	−0.5
[1-C_6_-Py][NTf_2_]	31.7 [36]	146.1 ± 1.8	145.9	0.2
[1-C_8_-Py][NTf_2_]	*29.0* ^d^	153.2 ± 2.4	153.0	0.2

^a^ Experimental data from Table 1. ^b^ Estimated using Equation (4). ^c^ Difference between columns 3 and 4. ^d^ Values have been derived from the chain-length dependence in Table 6.

**Table 8 molecules-27-02321-t008:** Experimental values of the surface tension *σ*_298_(exp) for the [1-C_n_-Py][NTf_2_] and [Me-1-C_n_-Py][NTf_2_] series available in the literature and the correlation of the vaporization enthalpies $\Delta_{l}^{g}H_{m}^{o}$(298.15 K) with the surface tension.

Ionic Liquid	*σ*_298_(exp)	$\Delta_{l}^{g}H_{m}^{o}(298.15\;K{)}_{\exp}{\;}^{a}$	$\Delta_{l}^{g}H_{m}^{o}(298.15\;K{)}_{\mathrm{calc}}{\;}^{b}$	Δ ^c^
	mN⋅m^−1^	kJ⋅mol^−1^	kJ⋅mol^−1^	
[1-C_2_-Py][NTf_2_]	37.4 [33]	131.7 ± 1.4	130.2	1.5
[1-C_3_-Py][NTf_2_]	35.4 [34]	135.4 ± 1.5	135.7	−0.3
[1-C_4_-Py][NTf_2_]	34.8 [35]	137.8 ± 1.4	137.3	0.5
[1-C_5_-Py][NTf_2_]	33.1 [Table 6]	141.7 ± 1.8	141.9	−0.2
[1-C_6_-Py][NTf_2_]	31.7 [36]	146.1 ± 1.8	145.7	0.4
[1-C_8_-Py][NTf_2_]	29.0 [Table 6]	153.2 ± 2.4	153.1	0.1
[2-Me-1-C_2_-Py][NTf_2_]	38.5 [37]	-	127.2	-
[2-Me-1-C_3_-Py][NTf_2_]	36.9 [37]	-	131.6	-
[3-Me-1-C_3_-Py][NTf_2_]	35.8 [38]	132.4 ± 1.6	134.6	−2.2
[4-Me-1-C_3_-Py][NTf_2_]	35.2 [34]	-	136.2	-
[2-Me-1-C_4_-Py][NTf_2_]	36.3 [35]	-	133.2	-
[3-Me-1-C_4_-Py][NTf_2_]	35.5 [35]	-	135.4	-
[4-Me-1-C_4_-Py][NTf_2_]	35.0 [35]	-	136.8	-

^a^ Experimental data from Table 1 and Table 2. ^b^ Estimated from Equation (5), the assessed expanded uncertainty of ± 2.0 kJ⋅mol^−1^ (with *k* = 2 and confidence level 0.95). ^c^ Difference between columns 3 and 4.

**Table 9 molecules-27-02321-t009:** Experimental values of the surface tension *σ*_298_(exp) for the [CN-1-C_n_-Py][NTf_2_] series available in the literature and the correlation of the vaporization enthalpies $\Delta_{l}^{g}H_{m}^{o}$(298.15 K) with the surface tension.

Ionic Liquid	*σ*_298_(exp)	$\Delta_{l}^{g}H_{m}^{o}(298.15\;K{)}_{\exp}{\;}^{a}$	$\Delta_{l}^{g}H_{m}^{o}(298.15\;K{)}_{\mathrm{calc}}{\;}^{b}$	Δ ^c^
	mN⋅m^−1^	kJ⋅mol^−1^	kJ⋅mol^−1^	
[3-CN-1-C_4_-Py][NTf_2_]	32.00 [35]	151.0 ± 2.1	151.5	−0.5
[3-CN-1-C_6_-Py][NTf_2_]	29.37 [35]	162.1 ± 2.4	162.4	−0.3
[4-CN-1-C_6_-Py][NTf_2_]	30.60 [35]	158.4 ± 2.3	157.3	1.1
[3-CN-1-C_8_-Py][NTf_2_]	28.65 [35]	165.3 ± 2.7	165.4	−0.1

^a^ Experimental data from Table 1. ^b^ Estimated from Equation (6), with the assessed expanded uncertainty of ± 2.0 kJ⋅mol^−1^ (with *k* = 2 and confidence level 0.95). ^c^ Difference between columns 3 and 4.

**Table 10 molecules-27-02321-t010:** Specific “transfer” contribution, $\Delta_{l}^{g}H_{m}^{o}$ **(H→****R_1_)** derived [13] from vaporization enthalpies of substituted pyridines or quinolines. R = Me, CN, or Et (at 298.15 K in kJ·mol^−1^) ^a^.

R_1_	$\Delta_{l}^{g}H_{m}^{o}(H\to R_{1}){\;}^{b}$
2-methyl-	2.3 ± 0.2
3-methyl-	4.4 ± 0.3
4-methyl-	4.7 ± 0.3
2-cyano-	18.4 ± 0.4
3-cyano-	15.6 ± 0.7
4-cyano-	13.8 ± 0.8
2-ethyl-	5.7 ± 0.2 ^b^

^a^ Uncertainties are expanded uncertainties (0.95 level of confidence, k = 2). ^b^ Calculated as the difference between $\Delta_{l}^{g}H_{m}^{o}$(298.15 K) = 45.9 ± 0.4 kJ·mol^−1^ for 2-ethyl-pyridine [13] and $\Delta_{l}^{g}H_{m}^{o}$(298.15 K) = 40.2 ± 0.2 kJ·mol^−1^ for pyridine [43].

**Table 11 molecules-27-02321-t011:** Calculation of the vaporization enthalpies, $\Delta_{l}^{g}H_{m}^{o}$, of alkyl- and cyano-substituted pyridinium-based ILs using the “centerpiece approach” (at 298.15 K in kJ·mol^−1^) ^a^.

IL	$\Delta_{l}^{g}H_{m}^{o}(H\to R_{1}){\;}^{b}$	$\Delta_{l}^{g}H_{m}^{o}(\mathrm{CP}){\;}^{c}$	$\Delta_{l}^{g}H_{m}^{o}{}_{\left(\mathrm{CP}\right)}{\;}^{d}$	$\Delta_{l}^{g}H_{m}^{o}{}_{\left(\exp\right)}$	Δ ^e^
[3-Me-1-C_3_-Py][NTf_2_]	4.4 ± 0.3	135.4 ± 1.5	138.9 ± 1.6	132.4 ± 1.6 [Table 1]	−6.5 ± 2.3
[2-Me-1-C_2_-Py][NTf_2_]	2.3 ± 0.2	131.7 ± 1.4	134.0 ± 2.4	127.2 ± 2.0 [Table 1]	−6.8 ± 3.2
[2-Me-1-C_3_-Py][NTf_2_]	2.3 ± 0.2	135.4 ± 1.5	137.7 ± 2.5	131.6 ± 2.0 [Table 7]	−6.1 ± 3.2
[3-Me-1-C_3_-Py][NTf_2_]	4.4 ± 0.3	135.4 ± 1.5	139.8 ± 2.5	134.6 ± 2.0 [Table 7]	−5.2 ± 3.2
[4-Me-1-C_3_-Py][NTf_2_]	4.7 ± 0.3	135.4 ± 1.5	140.1 ± 2.5	136.2 ± 2.0 [Table 7]	−3.9 ± 3.2
[2-Me-1-C_4_-Py][NTf_2_]	2.3 ± 0.2	137.8 ± 1.4	140.1 ± 2.4	133.2 ± 2.0 [Table 7]	−6.9 ± 3.2
[3-Me-1-C_4_-Py][NTf_2_]	4.4 ± 0.3	137.8 ± 1.4	142.2 ± 2.5	135.4 ± 2.0 [Table 7]	−6.8 ± 3.2
[4-Me-1-C_4_-Py][NTf_2_]	4.7 ± 0.3	137.8 ± 1.4	142.5 ± 2.5	136.8 ± 2.0 [Table 7]	−5.7 ± 3.2
[3-CN-1-C_4_-Py][NTf_2_]	15.6 ± 0.7	137.8 ± 1.4	153.4 ± 2.6	151.2 ± 2.1 [Table 1]	−2.2 ± 3.4
[3-CN-1-1-C_6_-Py][NTf_2_]	15.6 ± 0.7	146.1 ± 1.8	161.7 ± 3.1	162.1 ± 2.4 [Table 1]	0.4 ± 3.9
[4-CN-1-C_6_-Py][NTf_2_]	13.8 ± 0.8	146.1 ± 1.8	159.9 ± 3.0	158.4 ± 2.3 [Table 1]	−1.5 ± 3.8
[3-CN-1-C_8_-Py][NTf_2_]	15.6 ± 0.7	153.2 ± 2.4	168.8 ± 3.7	165.3 ± 2.7 [Table 1]	−3.5 ± 4.6
[2-Et-1-C_3_-Py][NTf_2_]	5.7 ± 0.2	135.4 ± 1.5	141.1 ± 3.6	137.2 ± 3.3 [Table 3]	−3.9 ± 4.9
[2-Et-1-C_4_-Py][NTf_2_]	5.7 ± 0.2	137.8 ± 1.4	143.5 ± 3.6	138.7 ± 3.3 [Table 3]	−4.8 ± 4.9
[2-Et-1-C_5_-Py][NTf_2_]	5.7 ± 0.2	141.7 ± 1.8	147.4 ± 4.2	145.1 ± 3.8 [Table 3]	−2.3 ± 5.7
[2-Et-1-C_6_-Py][NTf_2_]	5.7 ± 0.2	146.1 ± 1.8	151.8 ± 4.1	146.5 ± 3.7 [Table 3]	−5.3 ± 5.5
[2-Et-1-C_7_-Py][NTf_2_]	5.7 ± 0.2	149.4 ± 2.0 ^e^	155.1 ± 5.2	150.7 ± 4.8 [Table 3]	−4.4 ± 7.1
[2-Et-1-C_8_-Py][NTf_2_]	5.7 ± 0.2	153.2 ± 2.4	158.9 ± 4.9	158.4 ± 4.3 [Table 3]	−0.5 ± 6.5
[3-Me-1-C_4_-Py][BF_4_]	4.4 ± 0.3	149.9 ± 2.3 [31]	154.3 ± 3.3	149.5 ± 2.3 [31]	−4.8 ± 4.0
[4-Me-1-C_4_-Py][BF_4_]	4.7 ± 0.3	149.9 ± 2.3 [31]	154.6 ± 3.0	148.9 ± 2.1 [31]	−5.7 ± 3.8
				**average:**	−**4.9 ± 0.8** ^f^

^a^ Uncertainties of the vaporization enthalpy ($\Delta_{l}^{g}H_{m}^{o}$) are the expanded uncertainties (0.95 level of confidence, k = 2). ^b^ From Table 9. ^c^ Enthalpies of vaporization of the “centerpiece” molecules from Table 1, Table 2 and Table 3. ^d^ Calculated as the sum of columns 2 and 3. ^e^ Calculated as the difference of columns 5 and 4. ^f^ Weighted mean value (the uncertainty was taken as the weighing factor).

## Data Availability

The data supporting the reported results are given in the text and in the Appendix A.

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
