# Peer review of "Aprotic Ionic Liquids: A Framework for Predicting Vaporization Thermodynamics"

_molecules, 2022, doi:10.3390/molecules27072321_

Round 1

Reviewer 1 Report

Accept the manuscript in the current form

Author Response

THANK YOU VERY MUCH  FOR the POSITIVE OPINION!

Reviewer 2 Report

Review Comments on Manuscript molecules-1636436

Title: “Aprotic Ionic liquids: framework for predicting vaporization thermodynamics”

Author(s): Sergey P. Verevkin*, Dzmitry H. Zaitsau, Ralf Ludwig

Comments:

This manuscript mainly reports the vaporization thermodynamics of five pyridinium-based ionic liquids (ILs) by using a quartz crystal microbalance. According to the vapor pressure-temperature dependence, the vaporization enthalpies of these ILs were derived. Moreover, the vaporization enthalpies of the pyridinium-based ILs available in the literature were collected and uniformly adjusted to the reference temperature T = 298.15 K. The consistent sets of evaluated vaporization enthalpies were used to develop the “centerpiece” based group-additivity method for predicting enthalpies of vaporization of ionic compounds. However, the innovation and research significance of this manuscript are not outstanding enough. Hence, I do not recommend publication of this manuscript in Molecules based on the following comments.

  1. Page 2, line 54, the structural formulas of samples should be corrected to a ball-and-stick model in Fig. 1.
  2. Page 2, line 77, the purification method of samples and standard uncertainty for final mole fraction purity should be given in Table S1.
  3. The mass fraction of water in pure ILs should be given.
  4. The 1H NMR and 13C NMR spectrums of all samples should be given.
  5. The model, measuring principle, correction method and measurement accuracy of all measuring instruments should be given.
  6. Structures and abbreviations of samples should be summarized in a table.
  7. The surface tension should be expressed by γ in this paper.
  8. As shown in Table 8, for [Me-1-Cn-Py][NTf2], only the value of [3-Me-1-C3-Py][NTf2], and the differences between experimental and estimated values exceed 2 kJ·mol-1, it’s not very convincing.
  9. The format of references should be consistent as required.

Author Response

(The authors gave the same response as above.)

Reviewer 3 Report

The article titled “Aprotic Ionic liquids: framework for predicting vaporization thermodynamics” by Verevkin et al. is very interesting and well-designed piece of work. It offers new experimental data related to vaporization thermodynamics of selected ionic liquids with comprehensive data for the set of six titled ionic liquids. Authors used unique experimental setup for measurements of essential physicochemical properties necessary for precise description of molality of studied systems. Besides some theoretical investigations were provided for deeper insight of collected data. Conclusion are sound and proposed structure–property correlations are efficient diagnostic tools for other systems not studied so far. Hence the paper is publishable after few minor revisions mainly of an editorial and technical nature.

In the introduction the actual description of the approach is not clearly written. The idea graphically presented in Fig.2 and particularly meaning of ∆lg?mo (H→R1) is not univocally stated. Please help readers understanding this without deep referring to literature.

I do not understand what is written in lines 159-164. Why there are two distinct methodologies in context of COSMOtherm ? This software is just realization of COSMO-RS computations. Besides why incorrect citation 24 related to COSMO-RS? This is not acceptable and citations should address to original Klamt papers. What does it mean modified COSMOtherm? What version is used (the current one is labeled as version 21).

I do not find graphical visualization of the “centerpiece” approach as very appealing and related to methodology per se. Also the term “centerpiece” is misfortune. This is simply additive approach.

I recommend for publication with minor revision.

Author Response

(The authors gave the same response as above.)

Reviewer 4 Report

The manuscript entitled “Aprotic ionic liquids: framework for predicting
vaporization thermodynamics” reports an approach to predict the
vaporization enthalpies of pyridinium based ionic liquids and the contribution of
different groups attached to the pyridinium ring. The experimental data and
calculated values of vaporization enthalpies were compared to the literature
results for analogue structures and the results showed a good correlation. The
method certainly can be amplified, but it represents a considerable advance in
prediction of ionic liquids vaporization enthalpies, specially the pyridinium
derivatives. The language of the manuscript does not present any issue, with
just a few corrections needed. The imagens and tables have good resolution
and show the information in a proper way. Thus, I recommend this
manuscript to be accepted to publication in the Molecules journal, if the
authors can satisfactorily address the following comments:

1. The title of the manuscript. Correct the capital letter in “ionic liquids”.
2. Introduction, fourth sentence, lines 29, 30, 31. Please, rewrite the
sentence in a clearer way. It sounds a little confuse and does not provide
the true emphasis in the ionic liquids.
3. The authors should emphasize the reason for choosing the pyridinium
ILs over many other cations available, and why use the [NTf 2 ] anion.
4. Introduction, line 74. It’s quite incorrect to refer to the ILs used in the
work as “new”, since the six compounds introduced are commercial or
already reported.
5. Table 1, entry 4. Correct the name of the compound.
6. Figure 3. The image is a little confuse when attributed to the concept of
the centerpiece.
7. Section 3.4, line 305. I would suggest to not use the “meta-position” term,
maintain the numerical position as previously.
8. The conclusion is too embracing. The method is in fact a good
approximation and a useful tool, but, more complex substituents in the
pyridinium ring, and different anions, as organic molecular anions, can
lead to complex systems with more variables.

Author Response

(The authors gave the same response as above.)

Round 2

Reviewer 2 Report

This manuscript has been greatly improved after careful revision, I agree that this manuscript will be published in  Molecules.